# RNA Sequencing Reveals Unique Transcriptomic Signatures of the Thyroid in a Murine Lung Cancer Model Treated with PD-1 and PD-L1 Antibodies

**DOI:** 10.3390/ijms241310526

**Published:** 2023-06-23

**Authors:** Rena Pollack, Joshua Stokar, Natan Lishinsky, Irina Gurt, Naomi Kaisar-Iluz, Merav E. Shaul, Zvi G. Fridlender, Rivka Dresner-Pollak

**Affiliations:** 1Department of Endocrinology and Metabolism, Hadassah Medical Center, Jerusalem 91120, Israel; 2Faculty of Medicine, Hebrew University of Jerusalem, Jerusalem 91120, Israel; 3Institute of Pulmonary Medicine, Hadassah Medical Center, Jerusalem 91120, Israel

**Keywords:** immune checkpoint inhibitors, anti PD-1, thyroid, immune related adverse events, transcriptome

## Abstract

Immune checkpoint inhibitors (ICI) are commonly associated with thyroid immune-related adverse events, yet the mechanism has not been fully elucidated. We aimed to further explore the mechanism of ICI-induced thyroid dysfunction by assessing changes induced in the thyroid transcriptome by ICI treatment (αPD-1/αPD-L1) in a lung cancer murine model. RNA-sequencing of thyroid tissues revealed 952 differentially expressed genes (DEGs) with αPD-1 treatment (|fold-change| ≥1.8, FDR < 0.05). Only 35 DEG were identified with αPD-L1, and we therefore focused on the αPD-1 group alone. Ingenuity Pathway Analysis revealed that of 952 DEGs with αPD-1 treatment, 362 were associated with functions of cell death and survival, with predicated activation of pathways for apoptosis and necrosis (Z = 2.89 and Z = 3.21, respectively) and negative activation of pathways for cell viability and cell survival (Z = −6.22 and Z = −6.45, respectively). Compared to previously published datasets of interleukin-1β and interferon γ-treated human thyroid cells, apoptosis pathways were similarly activated. However, unique changes related to organ inflammation and upstream regulation by cytokines were observed. Our data suggest that there are unique changes in gene expression in the thyroid associated with αPD-1 therapy. ICI-induced thyroid dysfunction may be mediated by increased tissue apoptosis resulting in destructive thyroiditis.

## 1. Introduction

Immune checkpoint inhibitors (ICIs) have transformed the care of cancer patients, successfully improving overall survival in a wide range of malignancies. It is now understood that a key mechanism in the development of cancer results from the failure of the host immune system to control tumor progression, otherwise known as “cancer immune escape”. Anti-cancer immunotherapy aims to improve the ability to immunologically reject the tumor by generating an adequate immune response, breaking tumor-induced immune tolerance [1].

Current agents are monoclonal antibodies targeting either cytotoxic T-lymphocyte antigen 4 (CTLA-4) programmed cell death 1 (PD-1) or its ligand PD-L1 to potentiate anti-tumor immune responses. They have now become first-line therapy for metastatic melanoma, renal cell carcinoma, non-small cell carcinoma, and head and neck cancers and are expected to change the current conventional interventions for multiple other cancers. Unfortunately, these agents are associated with a wide range of toxicities, termed immune-related adverse events (irAEs) [2]. Endocrinopathies are among the most common irAEs and include hypophysitis, adrenal insufficiency, type 1 diabetes, and most frequently, thyroid dysfunction [3]. Real-world studies have reported thyroid dysfunction in up to 50 percent of patients receiving ICIs [4,5,6,7]. While thyroid irAEs are associated with increased immune toxicity in other organ systems, they are also associated with improved progression-free and overall survival [8,9,10].

The underlying mechanism of ICI-induced thyroid dysfunction remains elusive [11]. Several studies have suggested that ICI-induced thyroid dysfunction results from the unleashing of cytotoxic T cells against thyroid antigens [12,13,14]. This, in turn, leads to activation of the humoral response and secondary antibody production [15]. Thyroid autoantibodies, particularly targeting thyroglobulin (anti-Tg), thyroid peroxidase (anti-TPO), and occasionally, the thyroid stimulating hormone receptor (TRAb) have been detected. However, the pathogenesis is likely multifactorial as evidenced by the differences in clinical presentation, antibody positivity, genetic factors, and outcome data. It has been suggested that patients who experience irAEs may have a heightened immunological state prior to ICI initiation. Indeed, preexisting thyroid autoantibodies or higher BMI, suggestive of a proinflammatory metabolic state, have been implicated in thyroid irAEs [5,11]. A recent study identified higher levels of activated CD4^+^ memory T-cells and increased diversity in the T-cell receptor in peripheral blood as pretreatment predictors of ICI-induced irAEs, irrespective of the organ involved [8]. These same factors were identified in patients with autoimmune disease relative to healthy controls, suggesting that irAEs may result from unmasking of latent autoimmunity which predates ICI-initiation. The high prevalence of thyroid antibodies in the general population may explain why thyroid irAEs are common [11,12,13].

To further explore the molecular mechanism of ICI-induced thyroid dysfunction, we profiled changes in transcription induced by monoclonal antibodies targeting programmed cell death-1 (αPD-1) and programmed cell death-ligand 1 (αPD-L1) in a non-small cell lung cancer (NSCLC) murine model using RNA sequencing. Bioinformatic tools were used to compare transcriptional changes in our model to previously published transcription datasets of immune-mediated thyroid dysfunction.

## 2. Results

### 2.1. αPD-1 and αPD-L1 Treatment Induced an Anti-Tumor Response in Treated Mice

The overall study design is depicted in Figure 1A. Over the 7-day course of treatment, the relative tumor volume percentage significantly increased in control mice by 37.87% ± 19.16%, while it dramatically decreased by 36.21% ± 15.26%; *p* < 0.001 and 26.12% ± 11.39% in αPD-1 and αPDL-1-treated mice, respectively; *p* < 0.001 (Figure 1B).

### 2.2. αPD-1 Treatment Induced Unique Transcriptional Changes in the Thyroid

RNA-seq was performed in the thyroids of Lung KRas modified (LKRM) tumor-bearing control mice and in the αPD-1 and αPD-L1 treated mice. Principal component analysis (PCA) showed samples cluster by treatment type and not by technical batch (Figure 2A). A total of 952 differentially expressed genes (DEGs), including 265 upregulated and 688 downregulated, were identified with αPD-1 treatment, while only 35 genes were differentially expressed with αPD-L1 treatment (Figure 2B). Therefore, we chose to focus on the αPD-1 treated group alone. The spread of differently expressed genes (DEGs) by treatment group is seen in the volcano plot (Figure 2C). A heatmap of the RNA-seq data shows the overall separation between αPD-1 and vehicle-treated mice, demonstrating a differential transcriptional response to treatment (Figure 2D).

To further investigate the potential mechanisms of ICI-induced thyroid dysfunction, differentially expressed genes were uploaded to QUIAGEN IPA for pathways analysis. The leading canonical pathways found to be enriched for are listed in Figure 3A. DEGs included in the top canonical pathway are shown in Figure 3B. Categories for diseases and functions are shown in Figure 3C. Of the DEGs in the αPD-1 treatment group, 362 were associated with functions of cell death and survival, with a predicated activation of pathways for apoptosis and necrosis (Z = 2.89 and Z = 3.21, respectively) and negative activation of pathways for cell viability and cell survival (Z = −6.22 and Z = −6.45, respectively; Figure 3D).

### 2.3. Transcriptional Signature following αPD-1 Treatment Is Distinct when Compared with Other Forms of Immune Thyroid Dysfunction

To examine whether the transcriptional changes induced in the thyroid by αPD-1 converge with changes in biological pathways due to previously established forms of immune thyroid disease, we used the gene expression omnibus (GEO) to obtain transcriptomes of interleukin-1β-treated human thyroid cells, interferon gamma-treated human thyroid cells, and Graves’ disease mouse model thyroid tissues [14,15]. IPA’s analysis match feature then allowed us to create a uniform “analysis-to-analysis” comparison, circumventing differences in platforms, methodology, and species. This comparison reveled marked differences in pathway enrichment as well as in the predicted upstream regulators between the three types of immune thyroid disease (Figure 4), supporting the notion that ICI induced thyroid disease represents a novel form of immune thyroid disease with distinct underlying biological pathways and regulators. 

## 3. Discussion

In this study, we demonstrate that there are unique changes in gene expression in the murine thyroid associated with αPD-1 treatment. Interestingly, although both αPD-1 and αPDL-1 treatments showed an anti-tumor response, only treatment with αPD-1 induced significant transcriptional changes. This finding seems compatible with human studies showing a higher percentage of thyroid dysfunction associated with αPD-1 compared to that with αPD-L1 treatment [3,16,17].

As to be expected by the drug’s mechanism of action, the transcriptional changes in the thyroid induced by αPD-1 treatment involved pathways for cell-death and survival. Importantly, when comparing the changes in the thyroid transcriptional landscape induced by αPD-1 treatment in our model to those seen in a model of drug-induced thyroiditis and Graves’ disease, we found unique transcriptional features in the thyroid in αPD-1-treated mice. Most notable, was the striking difference in predicted up-stream cytokine regulators between the models. αPD-1 treatment resulted in negative activation of up-stream cytokine regulators in contrast to the other models. Interestingly, a recent study in humans found that in patients who developed thyroid dysfunction, ICI indeed led to a reduction in the serum levels of several cytokines [18].

The underlying mechanisms of ICI-induced thyroid disease are still not fully understood, and preclinical animal models remain challenging. Recent attempts include pre-immunization with human thyroglobulin and use of autoimmune prone mouse strains such as the NOD mouse [19,20]. In the first mouse model of αPD-1-induced thyroiditis, CBA/J immune-competent mice were immunized with human thyroglobulin (Tg) followed by treatment with αPD-1. Destructive thyroiditis was found in Tg-immunized mice but not in non-immunized mice. This study mimics ICI-induced thyroiditis in humans, as patients with underlying thyroid autoimmunity were shown to be more susceptible to the development of thyroid dysfunction when treated with ICI [8,11,21,22]. A key role for cytotoxic memory CD4^+^ T cells was demonstrated with no significant role for B cells. Interestingly, anti-Tg antibodies rather than anti-thyroid peroxidase antibodies seem to be more associated with ICI-induced thyroid dysfunction. Yet, this model does not explain the development of ICI-induced thyroiditis without underlying thyroid autoimmunity. In a recently published study on NOD mice, the administration of αPD-1 alone or in combination with anti-CTL4-Ab resulted in thyroidal activation of IL-17A ^+^T cells, while blocking IL-17A and TNF-α, an inducer of Th17 cells differentiation, effectively reduced thyroidal autoimmune infiltrates in ICI-treated non-tumor-bearing NOD mice. Moreover, when NOD mice were inoculated with melanoma or colon carcinoma tumor cells, increased ICI-induced thyroid Th17 and Tc17 was observed that was reduced with IL-17A neutralizing Ab. Importantly, IL-17A inhibition did not compromise the anti-tumor efficacy of ICI treatment. Of note, mild thyroidal autoimmune infiltrates were observed in the thyroid in young NOD mice even without any ICI therapy, illustrating the inherent tendency of this mouse strain to develop spontaneous autoimmunity. Interestingly, our data show down-regulation of the IL-17 signaling pathway compared to human thyroid cells treated with interleukin 1β + interferon γ and thyroid tissue derived from a murine model of Graves’ disease. Thus, the role of IL-17A^+^ T cells and IL-17A in ICI-induced thyroid disease and other irAEs needs to be further evaluated in additional mouse models and in humans.

This study has several strengths. To our knowledge, this is the first study to investigate the transcriptional landscape induced by αPD-1 in the thyroid in female mice inoculated with lung tumor cells using an unbiased approach of RNA sequencing and bioinformatic analyses comparing the transcriptional changes induced by αPD-1 and other forms of thyroid disease including drug-induced thyroiditis and autoimmune thyroiditis. Our data suggest unique pathways involved in ICI-induced thyroid disease with gene expression changes indicating activation of oxidative phosphorylation, cell necrosis, and cell death pathways. The changes observed in this study in multiple genes involved in oxidative phosphorylation require further exploration and may be related to the invasion of the thyroid by immune cells. Interestingly, activation of the oxidative phosphorylation pathway was recently reported in a subset population of CD8^+^ T lymphocytes infiltrating the tumor and in peripheral blood in melanoma patients, named CD8^+^ T_OXPHOS_. These CD8^+^T_OXPHOS_ had higher representation in ICI-resistant melanoma patients [23].

This study has several limitations. We did not measure thyroid hormones or thyroglobulin in the serum, nor did we perform histopathological evaluation of the thyroid to unravel the implications of αPD-1 administration on thyroid function or histology. Secondly, we did not include a group of αPD-1-treated tumor-free mice to understand the independent effects of αPD-1 on thyroid gene expression. Thirdly, thyroid tissue was not immediately perfused after euthanasia. Thus, contamination from circulating blood immune cells cannot be excluded. As we used bulk RNA-seq, we could not differentiate effects on different cell-types within the thyroid. Lastly, as in any animal study, the relevance of the results to humans needs to be assessed. Further research using thyroid tissue from αPD-1-treated humans and single cell RNA sequencing seem warranted.

## 4. Materials and Methods

### 4.1. Animal Model

C57BL/6 female mice, 5–6 weeks of age, 20–25 g, purchased from Harlan Laboratories (Jerusalem, Israel) were bred with 129/SVJ mice purchased from Jackson Laboratories, and only the first generation was used in experiments. Mice were housed under specific pathogen-free conditions at the Hebrew University School of Medicine Animal Resource Center. All experiments were executed in compliance with institutional guidelines and regulations, and the protocols were approved by the Animal Research Committee of the Hebrew University School of Medicine (approval number MD-17-15237-5). In all experiments, animals were euthanized before surgery by carbon dioxide inhalation.

### 4.2. Cell Line and Tumor Injection

LKRM (Lung K-Ras G12D mutant Modified) cells, a kind gift from Prof. Steven Albelda, University of Pennsylvania, Philadelphia, PA, USA, were cultured and maintained in DMEM media, supplemented with 10% heat-inactivated fetal bovine serum, 2 mM glutamine, 100 U/mL penicillin, and 100 μm/mL streptomycin (Biological Industries, Beit-Haemek, Israel) [24]. Cell cultures were maintained at 37 °C and 5% CO_2_. Cell cultures were regularly tested and maintained negative for mycoplasma contamination. B6/129 mice were injected into the flank with 1 × 10^6^ LKRM tumor cells suspended in 100 μL of PBS. Following tumor inoculation, tumor size was monitored every 1–3 days, and tumor volume was calculated using the formula volume = length × width^2^ × 3.14/6. After reaching a tumor size of approximately 200 mm^3^, LKRM tumor-bearing mice were injected IP with a volume of 100 μL containing 250 μg αPD-1, αPD-L1 antibodies (BioXCell, Lebanon, NH, USA), clone RMPI-14, clone 10F.9G2, respectively) or vehicle every 3 days for a total of 3 injections (day 0, 3, and 6). Tumor volume was measured every 3 days. Mice were sacrificed on day 7, and the thyroid glands were harvested and immediately frozen at −80 °C until analyzed.

### 4.3. RNA Extraction, Library Preparation, and RNA Sequencing

Total RNA from thyroid tissue was isolated using the RNeasy Plus Micro Kit (QIAGEN, Valencia, CA, USA) according to manufacturer’s instructions. RNA quantity and purity were confirmed with a Nanodrop spectrophotometer (Thermo Scientific, Wilmington, DE, USA). RNA samples with an RNA Integrity Number (RIN) of more than 6.5 confirmed by Agilent TapeStation system were sequenced (vehicle N = 3; αPD-1 N = 4; αPD-L1 N = 3).

Sequencing libraries were prepared using MARS-seq protocol [25]. Single-end reads were sequenced on 1 of an Illumina NovaSeq SP. The output was ~19 million reads per sample. Poly-A/T stretches and Illumina adapters were trimmed from the reads using cutadapt [26]; resulting reads shorter than 30 bp were discarded. Remaining reads were mapped onto 3′ UTR regions (1000 bases) of the M. musculus, mm10 genome according to Refseq annotations, using STAR [27] with EndToEnd option, and outFilterMismatchNoverLmax was set to 0.05. Deduplication was carried out by flagging all reads that were mapped to the same gene and had the same UMI. Counts for each gene were quantified using htseq-count [28], using the gtf above and corrected for UMI saturation. Differentially expressed genes were identified using DESeq2 [29] with the betaPrior, cooksCutoff, and independentFiltering parameters set to False. Raw *p* values were adjusted for multiple testing using the procedure of Benjamini and Hochberg. Pipeline was run using snakemake [30]. Genes were considered as differentially expressed if the adjusted *p* value was <0.05 and absolute fold-change was ≥1.8.

### 4.4. Pathway Analysis

Differentially expressed genes were uploaded into Qiagen’s Ingenuity Pathways Analysis (IPA) software (Qiagen, Hilden, Germany). IPA was used for subsequent bioinformatics analysis, which included canonical pathway analysis, disease and function, regulator effects, upstream regulators, and molecular networks. The analysis match feature was used to compare our dataset to two previously published datasets of immune thyroid dysfunction. The first dataset included interleukin-1β and interferon-gamma treated thyroid cell lines (GEO accession number GSE5054) [14]. In this study, primary thyroid cells were incubated with vehicle, 100 IU/mL IFN-gamma, 50 IU/mL IL1-beta, or a combination of both IFN-gamma and IL1-beta for 24 or 72 h. The experiment was repeated 5 times using thyroid cells from 5 different patients. One array was hybridized per sample (20 samples). The second dataset was a Graves’ disease mouse model (GEO accession number GSE39081) [15]. RNA sequencing was performed in thyroid tissues removed from CD40 over-expressing transgenic and wild type mice.

### 4.5. Statistical Analysis

Data are shown as mean ± standard error (SE) and analyzed using paired Student’s *t*-test to compare means of two groups and ANOVA for 3 groups. Normal distribution was confirmed through Shapiro–Wilk test. Statistical significance was calculated using GraphPad Prism for Windows Version 9 (GraphPad Software, La Jolla, CA, USA). Differences were considered significant when *p* < 0.05.

## 5. Conclusions

Overall, our results seem to support the hypothesis that α PD-1-induced thyroiditis is a unique form of thyroid disease. With the rapid increase in clinical indications for immunotherapy, thyroid dysfunction is expected to become a major clinical concern that will affect patients’ morbidity and quality of life. Moreover, the underlying mechanism of ICI-induced thyroiditis is of particular interest as its development was shown to be coupled with improved progression-free and total survival across different cancer types and in multiple studies [31]. Further research into the seemingly novel underlying mechanisms of ICI-induced thyroid dysfunction can lead to the identification of additional biomarkers for prediction of thyroid disease and other ICI-related irAEs as well as potential targets for its treatment and prevention, uncoupling the anti-tumor effects and increasing safety.

## Figures and Tables

**Figure 1 ijms-24-10526-f001:**
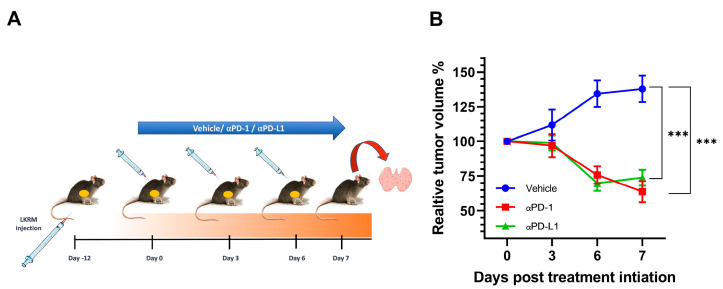
αPD-1 and αPD-L1 reduced tumor volume in Lung KRas modified (LKRM) tumor-bearing mice. (**A**) Overall study design. (**B**) Relative tumor volume over time (%); comparisons using one-way ANOVA with Holme–Sidak correction; *** *p* < 0.001 vs. vehicle; N = 3–4 mice per group.

**Figure 2 ijms-24-10526-f002:**
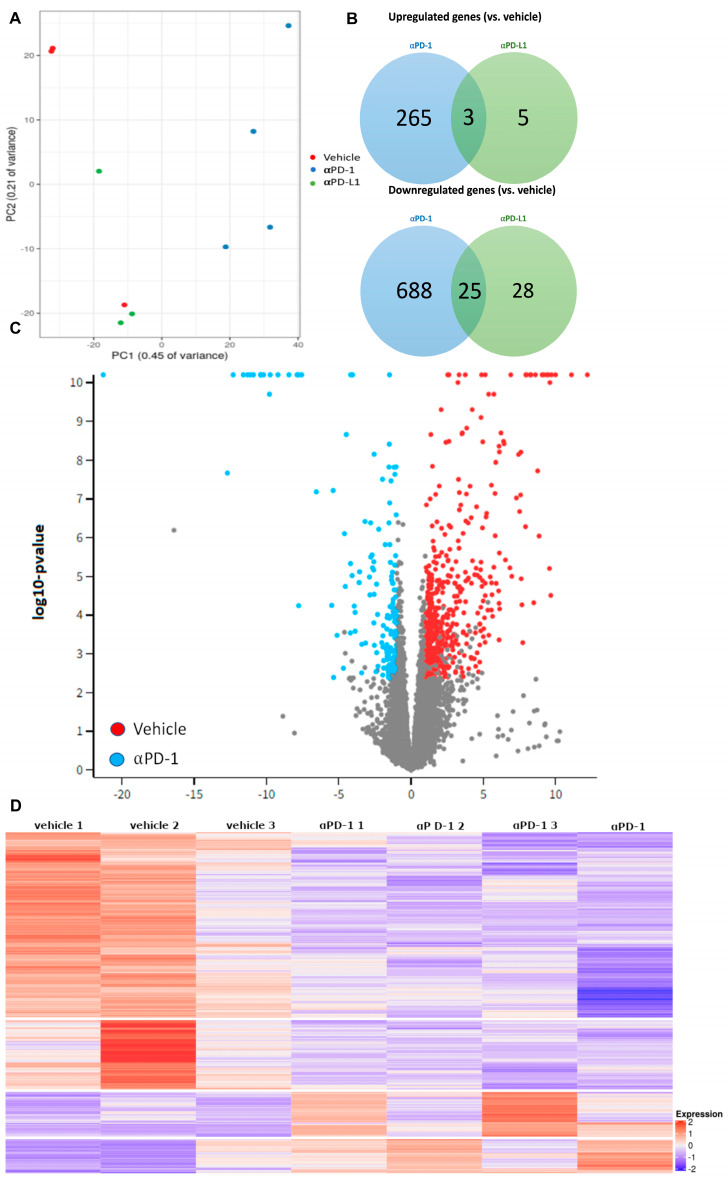
αPD-1 treatment induced changes in transcription in thyroid tissues. (**A**) Principal Component Analysis (PCA) based on top 1000 most variable genes. (**B**) Differently expressed genes (DEGs) vs. vehicle. (**C**) Volcano plot of DEGS for αPD-1 vs. vehicle. (**D**) Heatmap of individual gene expression. (**A**–**D**) N = 3 for vehicle and αPD-L1; N = 4 for αPD-1.

**Figure 3 ijms-24-10526-f003:**
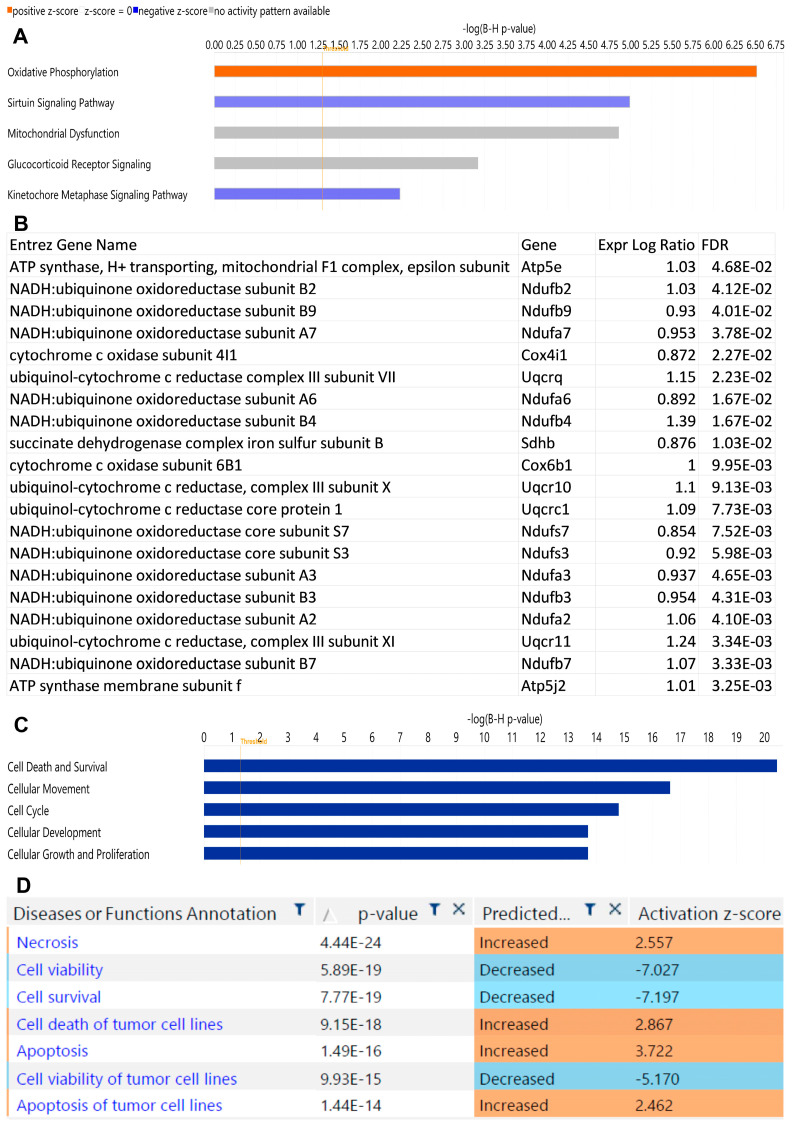
Ingenuity Pathway Analysis (IPA) for αPD-1 vs. vehicle. (**A**) Top canonical pathways. (**B**) List of DEGs for oxidative phosphorylation pathway. (**C**) Top changes in disease and functions. (**D**) Annotation of changes in cell death and survival category. (**A**–**D**) αPD-1 vs. vehicle; N = 3 for vehicle; N = 4 for αPD-1. Threshold at Bonferroni *p* = 0.05.

**Figure 4 ijms-24-10526-f004:**
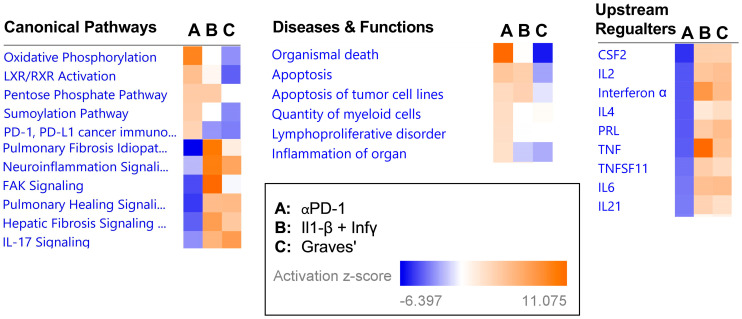
Comparison of pathway enrichment analyses of different immune thyroid diseases using previously published publicly available transcriptome datasets from the gene expression omnibus (GEO) and IPA “analysis match”. A. αPD-1 vs. vehicle; B. Interleukin 1β (Il1β) + Interferon γ (Inf γ) vs. vehicle (GEO GSE5054); C. Mouse model for Graves’ disease vs. control (GEO GSE39081).

## Data Availability

Not applicable.

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
