# Peer review of "RNA Sequencing Reveals Unique Transcriptomic Signatures of the Thyroid in a Murine Lung Cancer Model Treated with PD-1 and PD-L1 Antibodies"

_ijms, 2023, doi:10.3390/ijms241310526_

Round 1
Reviewer 1 Report
The manuscript ijms-2449325 entitled RNA Sequencing Reveals Unique Transcriptomic Signatures of the Thyroid in a Murine Lung Cancer Model Treated with PD-1 and PD-L1 Antibodies by Rena Pollack aimed to explore the mechanism of ICI-induced thyroid dysfunction by assessing changes induced in the thyroid tran-scriptome by ICI treatment (⍺PD-1/⍺PD-L1) in a lung cancer murine model.
The RNA-sequencing of thyroid tissues revealed 952 differentially expressed genes (DEGs) with ⍺PD-1 treatment (|fold-change| ≥1.8, FDR<0.05). Only 35 DEG were identified with ⍺PD-L1, and we therefore focused on the ⍺PD-1 group alone. Of 952 DEGs with ⍺PD-1 treatment, 362 were associated with functions of cell death and survival, with predicated activation of pathways for apoptosis and necrosis (Z=2.89 and Z=3.21, respectively) and negative activation of pathways for cell viability and cell survival (Z=-6.22 and Z=-6.45, respectively). The data suggest that there are unique changes in gene expression in the thyroid associated with ⍺PD-1 therapy. ICI-induced thyroid dysfunction may be mediated by increased tissue apoptosis resulting in de-structive thyroiditis.
The experimental design is clear
Results are significant and figure are informative
Discussion is consistent with the results.
Minor Comments
At the end of the discussion, a separated section could be dedicated as the conclusive remarks
English lenguage is appropriate
Author Response
Thank you for your comment. As recommended, a separate conclusion section has been added to end of the manuscript in accordance with the optional journal guidelines.
Reviewer 2 Report
The manuscript by Pollack and co-workers studies the changes on the transcriptional level in mice lung cancer model treated with immune checkpoint inhibitors (ICI). The study is interesting, however it cannot be accepted in the current form. Suggested changes are listed below.
Major remarks:
[1] A significant limitation of the study is the fact that no biochemical results were presented that would indicate changes in thyroid function in mice after ICI administration. Please explain why such tests have not been carried out.
[2] Please explain why the experiment was terminated 7 days after the first administration of ICI. On what basis was the experiment endpoint chosen (literature data?).
[3] I would suggest that the Authors add basic information in the Introduction, such as what ICI is, why they are used to treat cancer, in what types of cancer they are used, etc.
[4] The Introduction also lacks an explanation of what thyroid autoantibodies are, what antigens they recognize. The Reader should be familiar with these issues, especially since in the Discussion the Authors refer to specific thyroid autoantibodies (anti-Tg and anti-TPO). I also wonder if the levels of other thyroid autoantibodies (anti-NIS or anti-TSHR) also change in patients after ICI therapy?
[5] The mouse thyroid gland is extremely small. I wonder how the Authors were sure that they were actually taking tissue from the thyroid gland and not random tissue located nearby? Please explain.
[6] Figure 4 legend: I would suggest expanding this legend to better explain what we see in Figure 4. At the same time, it should be better noted that some of the data presented here have already been published.
[7] Results Section 2.3.: The section is too short. There is no explanation of how the results shown in Figure 4 should be interpreted
[8] Section 4.2.: Provide composition of medium in which LKRM cells were suspended for injection. The volume of LKRM cells suspension is also missing. Similarly, the volume of injected antibodies is not provided.
Minor remarks:
[1] Line 247: Reference is needed.
[2] References are needed in lines 105-107.
[3] Please explain ‘SE’ used in line 249.
[4] Line 75: Please explain ‘LKRM’ at the first mention in the main body of the manuscript.
[5] Line 91: Please add information that IPA is provided by Qiagen.
[6] Explain ‘IP’ used in line 209.
The manuscript by Pollack and co-workers studies the changes on the transcriptional level in mice lung cancer model treated with immune checkpoint inhibitors (ICI). The study is interesting, however it cannot be accepted in the current form. Suggested changes are listed below.
Major remarks:
[1] A significant limitation of the study is the fact that no biochemical results were presented that would indicate changes in thyroid function in mice after ICI administration. Please explain why such tests have not been carried out.
[2] Please explain why the experiment was terminated 7 days after the first administration of ICI. On what basis was the experiment endpoint chosen (literature data?).
[3] I would suggest that the Authors add basic information in the Introduction, such as what ICI is, why they are used to treat cancer, in what types of cancer they are used, etc.
[4] The Introduction also lacks an explanation of what thyroid autoantibodies are, what antigens they recognize. The Reader should be familiar with these issues, especially since in the Discussion the Authors refer to specific thyroid autoantibodies (anti-Tg and anti-TPO). I also wonder if the levels of other thyroid autoantibodies (anti-NIS or anti-TSHR) also change in patients after ICI therapy?
[5] The mouse thyroid gland is extremely small. I wonder how the Authors were sure that they were actually taking tissue from the thyroid gland and not random tissue located nearby? Please explain.
[6] Figure 4 legend: I would suggest expanding this legend to better explain what we see in Figure 4. At the same time, it should be better noted that some of the data presented here have already been published.
[7] Results Section 2.3.: The section is too short. There is no explanation of how the results shown in Figure 4 should be interpreted
[8] Section 4.2.: Provide composition of medium in which LKRM cells were suspended for injection. The volume of LKRM cells suspension is also missing. Similarly, the volume of injected antibodies is not provided.
Minor remarks:
[1] Line 247: Reference is needed.
[2] References are needed in lines 105-107.
[3] Please explain ‘SE’ used in line 249.
[4] Line 75: Please explain ‘LKRM’ at the first mention in the main body of the manuscript.
[5] Line 91: Please add information that IPA is provided by Qiagen.
[6] Explain ‘IP’ used in line 209.
Author Response
Response to Reviewer 2:
Major remarks:
[1] A significant limitation of the study is the fact that no biochemical results were presented that would indicate changes in thyroid function in mice after ICI administration. Please explain why such tests have not been carried out.
This is true and is mentioned in the methods as a limitation of our study. Unfortunately, the serum retrieved from the mice was insufficient for thyroid hormone analysis. However, we believe that our results are still of interest as we demonstrate that there are unique changes in gene expression in the murine thyroid associated with ⍺PD-1 treatment, irrespective of biochemical changes.
[2] Please explain why the experiment was terminated 7 days after the first administration of ICI. On what basis was the experiment endpoint chosen?
We have made the decision to terminate the study after 7 days from the first injection in order to strike a balance between treatment efficacy and ethical considerations. The rationale behind this decision is to ensure that we can observe a sufficient tumor response while also preventing the control mice from experiencing excessively large tumors, which would be considered unethical. This protocol has been previously published: Kaisar-Iluz, N.; Arpinati, L.; Shaul, M.E.; Mahroum, S.; Qaisi, M.; Tidhar, E.; Fridlender, Z.G. The Bilateral Interplay between Cancer Immunotherapies and Neutrophils' Phenotypes and Sub-Populations. Cells 2022, 11, doi:10.3390/cells11050783.
[3] I would suggest that the Authors add basic information in the Introduction, such as what ICI is, why they are used to treat cancer, in what types of cancer they are used, etc.
Thank you for this comment. This information has been added to the introduction.
[4] The Introduction also lacks an explanation of what thyroid autoantibodies are, what antigens they recognize. The Reader should be familiar with these issues, especially since in the Discussion the Authors refer to specific thyroid autoantibodies (anti-Tg and anti-TPO). I also wonder if the levels of other thyroid autoantibodies (anti-NIS or anti-TSHR) also change in patients after ICI therapy?
Thank you for your suggestion. This information in now briefly included in the introduction. Unfortunately, there is no data regarding levels of anti-NIS.
[5] The mouse thyroid gland is extremely small. I wonder how the Authors were sure that they were actually taking tissue from the thyroid gland and not random tissue located nearby? Please explain.
Thank you for this important comment. Indeed, the mouse thyroid is quite small. To ensure correct dissection of the thyroid, we had this procedure performed by experienced veterinary technicians from our animal facility. To validate the presence of thyroid tissue in the specimens we examined the expression levels of thyroglobulin (TG), as well as the presence of other thyroid specific genes such as Thyroid peroxidase (TPO), Solute Carrier Family 5 Member 5 (SLC5A5) and Paired box 8 (PAX8).
[6] Figure 4 legend: I would suggest expanding this legend to better explain what we see in Figure 4. At the same time, it should be better noted that some of the data presented here have already been published.
Thank you for this helpful suggestion. We have implemented it in the figure 4 legend.
[7] Results Section 2.3.: The section is too short. There is no explanation of how the results shown in Figure 4 should be interpreted.
Thank you for this helpful suggestion. We have expanded the results section 2.3 accordingly.
[8] Section 4.2.: Provide composition of medium in which LKRM cells were suspended for injection. The volume of LKRM cells suspension is also missing. Similarly, the volume of injected antibodies is not provided.
This information has now been added to the methods.
Minor remarks:
[1] Line 247: Reference is needed.
This reference is the same listed in the previous line (#15).
[2] References are needed in lines 105-107.
These have been added.
[3] Please explain ‘SE’ used in line 249.
The abbreviation has now been clarified.
[4] Line 75: Please explain ‘LKRM’ at the first mention in the main body of the manuscript.
This has been added.
[5] Line 91: Please add information that IPA is provided by Qiagen.
This has been added.
[6] Explain ‘IP’ used in line 209.
The abbreviation has now been explained.
Round 2
Reviewer 2 Report
The manuscript has been significantly improved. I would only suggest improving the resolution of some figures, especially Figs. 2 and 3. Please explain “LKMR” acronym in Figure 1 legend.
Typos:
[1] Line 165: Change “anti Tg” for “anti-Tg”.
[2] Lines 88 and 103: Change “vs” for “vs.”.
The manuscript has been significantly improved. I would only suggest improving the resolution of some figures, especially Figs. 2 and 3. Please explain “LKMR” acronym in Figure 1 legend.
Typos:
[1] Line 165: Change “anti Tg” for “anti-Tg”.
[2] Lines 88 and 103: Change “vs” for “vs.”.
Author Response
The manuscript has been significantly improved. I would only suggest improving the resolution of some figures, especially Figs. 2 and 3. Please explain “LKMR” acronym in Figure 1 legend.
# Thank you for these comments. The figures have been updated to higher resolution. The acronym has now been explained in the legend.
Typos:
[1] Line 165: Change “anti Tg” for “anti-Tg”.
[2] Lines 88 and 103: Change “vs” for “vs.”.
# These typos have now been corrected.